# Influence-Disentangled Federated Training: Learning Models That Are Easy to Unlearn

**Canran Xiao** [1]    **Qianyu Chen** [2]    **Liwei Hou** [3]

## Abstract

Federated learning increasingly faces deletion requests that require client-level unlearning without sacrificing model quality, yet a client's influence is often deeply entangled after many rounds of aggregation. We aim to make unlearning fast, stable, and predictable by reducing the gap to leave-one-out retraining under realistic heterogeneity. We propose Influence-Disentangled Federated Training (IDFT), which instruments standard FedAvg with training-time influence logging: each round's updates are decomposed into shared covarying directions and a client-separable residual trace, and an entanglement-aware shrinkage suppresses non-removable components. Deletion then becomes a single subtraction followed by a short anchored repair, yielding a stability-style characterization of retrain fidelity driven by the unremoved residual. Across representative benchmarks, IDFT consistently attains the lowest retrain gap (Avg. Gap) on multiple dataset–architecture pairs and improves the fidelity–cost frontier, matching retrain-level forgetting with substantially lower communication/compute than history-heavy baselines. These results suggest a practical pathway to unlearning-friendly federated systems by designing for removability during training rather than relying solely on post-hoc corrections.

## 1. Introduction

Federated learning (FL) enables collaborative model training across decentralized clients while keeping raw data local, making it a natural fit for privacy-sensitive applications such as mobile personalization, cross-silo analytics, and healthcare (Li et al., 2020; Kairouz et al., 2021; Xiao & Hou, 2026). However, emerging privacy regulations and user expectations increasingly demand the right to be forgotten after training, i.e., the ability to revoke a client's contribution and obtain a model indistinguishable from one trained without that client. Delivering such federated unlearning reliably is challenging because FL couples clients through iterative aggregation, and non-IID participation amplifies cross-client co-adaptation and interference (Romandini et al., 2024; Chen et al., 2024; Wang et al., 2022; Zhao et al., 2025).

Most federated unlearning methods are *post-hoc* approximations to leave-one-out retraining, either reusing training artifacts or running extra optimization at deletion time—e.g., update-history reconstruction (Liu et al., 2021), retained-client correction rounds (Halimi et al., 2022), or distillation/rapid retraining (Wu et al., 2022a; Liu et al., 2022; Fraboni et al., 2024). A persistent tension is that a client's influence has already *diffused* through repeated aggregation, so clean removal often needs substantial server state or strong assumptions and can damage shared knowledge, especially under non-IID and repeated deletions. Recent state-of-the-art methods improve efficiency and selectivity via parameter or representation interventions (Khalil et al., 2025; Zhong et al., 2025; Guo et al., 2025; Gu et al., 2024) and considers verifiability (Gao et al., 2024), but training-time mechanisms that make per-client influence *separable and cheaply removable*—and a predictive link between *entanglement* and *retrain fidelity*—remain underexplored.

> **This paper asks:**
>
> *Can we train federated models so that future client deletions become a controlled, low-cost operation rather than an expensive approximation to retraining?*

We propose to instrument FL training with an influence-disentangling view that separates broadly shared directions from client-specific residual influence and records a compact removable trace for each client, so that unlearning can be executed as a fast removal followed by a short, constrained repair to recover the leave-one-out solution.

---

[1]Shenzhen Campus of Sun Yat-sen University, Shenzhen, Guangdong, China [2]Nanyang Technological University, Singapore [3]Airon Tech, Changsha, Hunan, China. Correspondence to: Liwei Hou <houliwei@hnu.edu.cn>.

*Proceedings of the 43rd International Conference on Machine Learning*, Seoul, South Korea. PMLR 306, 2026. Copyright 2026 by the author(s).

Our contributions are as follows:

(i) We formalize federated unlearning difficulty as a consequence of cross-client influence entanglement, motivating training-time structure that preserves shared knowledge while keeping client-specific influence weakly coupled.

(ii) We introduce **Influence-Disentangled Federated Training (IDFT)**, a framework that produces client-separable removable influence traces during standard FL training, enabling deletion via fast removal plus lightweight repair.

(iii) We provide theory connecting retrain fidelity to the unremoved residual influence and analyze when the shared component is reliably recoverable; empirically, IDFT achieves a stronger fidelity–cost trade-off than competitive recent state-of-the-art methods across representative benchmarks.

## 2. Related Work

Federated unlearning has progressed from post-hoc approximations of leave-one-out retraining toward unlearning-aware training and representations that make future deletions fast and stable under client heterogeneity. Early machine-unlearning work emphasizes limiting retraining cost by structuring training artifacts (Bourtoule et al., 2021). In federated settings, FedEraser reuses stored historical updates to reconstruct a removal model (Liu et al., 2021), while client-erasure style methods perform deletion-time correction with additional federated optimization on retained clients (Halimi et al., 2022). Other lines accelerate recovery via distillation (Wu et al., 2022a), rapid retraining schedules (Liu et al., 2022), or sequentially informed updates (Fraboni et al., 2024). A common limitation is that the target client's influence is already entangled through aggregation: post-hoc corrections often require substantial server-side history, rely on extra assumptions (e.g., access to deleted data or checkpoints), or may disrupt shared knowledge under strong non-IID, leading to unstable fidelity.

Recent work targets stronger selectivity and operational efficiency by intervening in representations or parameters: NoT uses weight negation to break co-adaptation without heavy history (Khalil et al., 2025); FUSED achieves reversible unlearning via selective sparse adapters that overwrite sensitive knowledge while freezing the backbone (Zhong et al., 2025); FUCRT performs class-aware representation transformation for class-wise unlearning (Guo et al., 2025); and FedAU injects lightweight auxiliary modules during training to support efficient unlearning (Gu et al., 2024). Parallel efforts study verifiability and compliance (e.g., VeriFi) (Gao et al., 2024). While effective, these approaches typically do not yield an additively attributable per-client influence that can be removed by a single subtraction, and the mechanisms that govern retrain fidelity are often implicit. In contrast, IDFT makes removability explicit at training time: it factorizes

each round into a shared subspace and a client-separable residual, applies entanglement-aware shrinkage to suppress non-removable components, and logs a compact per-client influence bank—so deletion reduces to subtraction plus a short anchored repair, with a stability-style retrain-gap characterization.

## 3. Method

Influence-Disentangled Federated Training (IDFT) augments standard federated training with a training-time influence logging mechanism that makes future deletions cheap and stable. During training, the server runs vanilla FedAvg updates for the global model $w_t$, while simultaneously factorizing each round's client updates into (i) a shared subspace that captures cross-client covarying directions and (ii) a client-specific influence subspace that stores low-entanglement residual components in a small coefficient vector. Upon a deletion request from client $u$, IDFT produces an initial unlearned model by subtracting the stored influence of $u$ from the trained model, then runs a short constrained repair over retained clients to match the retrained-without-$u$ solution.

### 3.1. Federated Setup and Unlearning Target

There are $N$ clients indexed by $i \in [N]$, each holding dataset $D_i$ of size $n_i$, and total $n = \sum_{i=1}^{N} n_i$. The federated objective is

$$\min_{w \in \mathbb{R}^d} F(w) \triangleq \sum_{i=1}^{N} \frac{n_i}{n} f_i(w),$$

$$\text{where} \quad f_i(w) \triangleq \mathbb{E}_{\xi \sim D_i}\big[\ell(w;\xi)\big]. \tag{1}$$

At round $t$, the server samples a participating set $S_t \subset [N]$ and uses *round-normalized* aggregation weights $p_{t,i} \triangleq \frac{n_i}{\sum_{j \in S_t} n_j}$ for $i \in S_t$.

**Training update (FedAvg).** Each participating client returns an update $\Delta_{t,i} \in \mathbb{R}^d$, and the server updates

$$w_{t+1} \triangleq w_t + \sum_{i \in S_t} p_{t,i} \, \Delta_{t,i}. \tag{2}$$

**Unlearning target.** Given a deletion request from client $u$, let $w^{\backslash u}$ denote the model obtained by rerunning the same training protocol from scratch on $\{D_i\}_{i \neq u}$. IDFT outputs $w^{-u}$ and measures fidelity by

$$\text{Gap}(u) \triangleq \big\| w^{-u} - w^{\backslash u} \big\|_2. \tag{3}$$

### 3.2. Training-Time Influence Logging

3.2.1. CLIENT LOCAL UPDATE

IDFT keeps the optimization path of the global model identical to standard FedAvg to preserve accuracy and conver-

gence behavior, while extracting and storing removable client-specific influence signals during training.

In round $t$, client $i \in S_t$ initializes at $w_t$, runs $E$ steps of local SGD on $f_i$, and returns the model difference

$$\Delta_{t,i} \triangleq \theta_{t,i} - w_t \in \mathbb{R}^d, \tag{4}$$

where $\theta_{t,i}$ is the client's local model after the $E$ SGD steps. The server uses $\{\Delta_{t,i}\}_{i \in S_t}$ in Eq. (2) and simultaneously factorizes them for influence logging.

### 3.2.2. SHARED SUBSPACE EXTRACTION

Influences that strongly covary across clients are shared and tend to remain even after removing one client. Routing these directions into a shared subspace reduces cross-client entanglement of what is logged as removable influence.

The server forms the weighted update matrix

$$\Delta_t \triangleq \left[\sqrt{p_{t,i}}\Delta_{t,i}\right]_{i \in S_t} \in \mathbb{R}^{d \times |S_t|}, \tag{5}$$

computes its thin SVD $\Delta_t = U\Sigma V^\top$, and sets $U_t \in \mathbb{R}^{d \times r}$ as the first $r$ columns of $U$ (so $U_t^\top U_t = I_r$). Each update is decomposed by orthogonal projection:

$$\Delta_{t,i}^{\text{sh}} \triangleq U_t U_t^\top \Delta_{t,i}, \qquad r_{t,i} \triangleq (I - U_t U_t^\top)\Delta_{t,i}. \tag{6}$$

where $r$ is the shared-subspace rank, $I$ is the $d \times d$ identity matrix, $\Delta_{t,i}^{\text{sh}} \in \mathbb{R}^d$ is the shared component, and $r_{t,i} \in \mathbb{R}^d$ is the orthogonal residual used for influence logging.

### 3.2.3. CLIENT INFLUENCE SUBSPACE

To make future deletion a subtraction, IDFT stores each client's logged influence in a client-specific orthonormal subspace so the stored quantity is compact and separable across clients.

Each client $i$ is assigned a fixed orthonormal basis $V_i \in \mathbb{R}^{d \times s}$ with $V_i^\top V_i = I_s$. $V_i$ is generated deterministically from the client index $i$ by: (1) a public pseudorandom generator seeded by $i$ produces a matrix $G_i \in \mathbb{R}^{d \times s}$ with i.i.d. standard normal entries, and (2) $V_i$ is the $Q$ factor from the thin QR factorization of $G_i$. The server then projects the residual onto $\text{span}(V_i)$:

$$\tilde{v}_{t,i} \triangleq V_i V_i^\top r_{t,i} \in \mathbb{R}^d. \tag{7}$$

where $s$ is the influence-subspace dimension, and $\tilde{v}_{t,i}$ is the residual component representable in client $i$'s influence subspace.

### 3.2.4. ENTANGLEMENT METRIC AND SHRINKAGE

Not every projected influence is equally removable: (i) influences aligned with other clients reflect shared information; (ii) influences spread across many layers reflect

co-adaptation; (iii) large projection error indicates leakage outside the influence subspace. IDFT quantifies these effects with an entanglement score and shrinks what is stored.

Let the parameter vector be partitioned into $L$ blocks (layers), so any $x \in \mathbb{R}^d$ decomposes as $x = (x^{(1)}, \ldots, x^{(L)})$. Define the weighted mean $\bar{v}_t \triangleq \sum_{j \in S_t} p_{t,j}\tilde{v}_{t,j}$ and a small constant $\varepsilon > 0$. IDFT defines

$$\mathcal{E}_{t,i} \triangleq a\,\mathcal{E}_{t,i}^{\text{cov}} + b\,\mathcal{E}_{t,i}^{\text{coa}} + (1 - a - b)\,\mathcal{E}_{t,i}^{\text{leak}}, \tag{8}$$

where

$$\mathcal{E}_{t,i}^{\text{cov}} \triangleq \frac{1}{1 - p_{t,i}} \sum_{j \in S_t \setminus \{i\}} p_{t,j} \times$$
$$\frac{\left|\langle \tilde{v}_{t,i} - \bar{v}_t,\ \tilde{v}_{t,j} - \bar{v}_t \rangle\right|}{\left(\|\tilde{v}_{t,i} - \bar{v}_t\|_2 + \varepsilon\right)\left(\|\tilde{v}_{t,j} - \bar{v}_t\|_2 + \varepsilon\right)}, \tag{9}$$

$$\mathcal{E}_{t,i}^{\text{leak}} \triangleq \frac{\|r_{t,i} - \tilde{v}_{t,i}\|_2^2}{\|r_{t,i}\|_2^2 + \varepsilon}, \tag{10}$$

$$\mathcal{E}_{t,i}^{\text{coa}} \triangleq -\sum_{\ell=1}^{L} q_{t,i}^{(\ell)} \log\left(q_{t,i}^{(\ell)} + \varepsilon\right), \quad q_{t,i}^{(\ell)} \triangleq \frac{\|\tilde{v}_{t,i}^{(\ell)}\|_2^2}{\|\tilde{v}_{t,i}\|_2^2 + \varepsilon}. \tag{11}$$

where $a, b \in [0, 1]$ with $a + b \leq 1$ weight the three terms; $\mathcal{E}_{t,i}^{\text{cov}}$ is a normalized cross-client covariance score (centered at $\bar{v}_t$); $\mathcal{E}_{t,i}^{\text{leak}}$ measures how much residual energy lies outside $\text{span}(V_i)$; $\mathcal{E}_{t,i}^{\text{coa}}$ is the entropy of the layer-energy distribution $q_{t,i}^{(\ell)}$, where $\tilde{v}_{t,i}^{(\ell)}$ denotes the block corresponding to layer $\ell$.

IDFT converts entanglement into a shrinkage factor and updates a compact influence coefficient:

$$\begin{aligned} s_{t,i} &\triangleq \frac{1}{1 + \lambda\mathcal{E}_{t,i}}, \\ z_{t+1,i} &\triangleq z_{t,i} + p_{t,i}\,s_{t,i}\,V_i^\top r_{t,i}, \\ \phi_{t,i} &\triangleq V_i z_{t,i}. \end{aligned} \tag{12}$$

where $\lambda \geq 0$ controls shrinkage strength; $z_{t,i} \in \mathbb{R}^s$ is client $i$'s stored influence coefficient at round $t$ (initialized as $z_{0,i} = 0$); $\phi_{t,i} \in \mathbb{R}^d$ is the stored influence vector used for deletion. For any $i \notin S_t$, the server sets $z_{t+1,i} \triangleq z_{t,i}$.

**Summary of Training Procedure.** The training model trajectory $\{w_t\}$ remains easy to interpret and exactly matches a standard FedAvg baseline, while the influence bank $\{z_{t,i}\}$ is built online during training to support future deletions.

For each round $t = 0, \ldots, T-1$, IDFT executes: **(i)** sample $S_t$ and broadcast $w_t$; **(ii)** collect client updates $\{\Delta_{t,i}\}_{i \in S_t}$; **(iii)** update the global model by FedAvg using Eq. (2); **(iv)** compute the shared basis $U_t$ from the thin SVD of the weighted update matrix $\Delta_t = [\sqrt{p_{t,i}}\Delta_{t,i}]_{i \in S_t}$ and obtain residuals $r_{t,i}$ via Eq. (6); **(v)** project each residual onto

the client influence subspace to obtain $\tilde{v}_{t,i}$ via Eq. (7) and compute the weighted mean $\bar{v}_t = \sum_{j \in S_t} p_{t,j} \tilde{v}_{t,j}$; **(vi)** compute the entanglement score $\mathcal{E}_{t,i}$ using Eq. (8) together with Eqs. (9)–(11); **(vii)** compute shrinkage and update the influence coefficients by Eq. (12). For any $i \notin S_t$, the server keeps $z_{t+1,i} \triangleq z_{t,i}$.

### 3.3. Deletion-Time Unlearning

**Step 1: Fast Remove by Subtraction.** Because the influence bank stores a per-client separable component $\phi_{T,u}$, deletion starts with a single subtraction. At the end of training (round $T$), the server has $w_T$ and stored influence $\phi_{T,u} = V_u z_{T,u}$. IDFT forms the initial unlearned model

$$w^{(-u,0)} \triangleq w_T - \phi_{T,u}. \quad (13)$$

where $w^{(-u,0)}$ is the subtraction-initialized model before repair.

**Step 2: Constrained Repair on Retained Clients.** Subtraction removes the low-entanglement part of client $u$'s influence. The remaining discrepancy to $w^{\backslash u}$ comes from shared and entangled effects and is corrected by a short repair that stays close to $w^{(-u,0)}$.

IDFT runs $R$ repair rounds ($R \ll T$) on clients $[N] \setminus \{u\}$ to minimize the anchored objective

$$
\begin{aligned}
&\min_{w \in \mathbb{R}^d} F_{-u}(w) + \frac{\beta}{2} \|w - w^{(-u,0)}\|_2^2, \\
&\text{with } F_{-u}(w) \triangleq \sum_{i \neq u} \frac{n_i}{\sum_{j \neq u} n_j} f_i(w),
\end{aligned}
\quad (14)
$$

where $\beta > 0$ enforces a trust region around $w^{(-u,0)}$. Each repair round uses the same local-update plus aggregation pattern as Eq. (2), applied to the per-client proximal loss $f_i(w) + \frac{\beta}{2} \|w - w^{(-u,0)}\|_2^2$ for each retained client $i$.

Algorithms 1–2 describe the complete IDFT algorithm.

## 4. Theory

### 4.1. Unlearning Fidelity via Stability/Sensitivity

We provide a stability-style retrain-gap bound for IDFT. The analysis conditions on the realized participation sequence $\{S_t\}_{t=0}^{T-1}$ and studies a standard FedSGD surrogate (one local step), which yields a clean contractive recursion and is widely used in theoretical analyses of federated optimization.

**Per-round retained objective.** For a fixed deletion request from client $u$, define the retained (leave-$u$-out) per-round

---

**Algorithm 1** IDFT Training

1: Initialize $w_0 \in \mathbb{R}^d$; set $z_{0,i} \leftarrow 0 \in \mathbb{R}^s$ for all $i \in [N]$; fix $V_i$ by deterministic QR seeded with $i$
2: **for** $t = 0, 1, \ldots, T-1$ **do**
3:     Sample $S_t$; set $p_{t,i} \leftarrow \frac{n_i}{\sum_{j \in S_t} n_j}$ for $i \in S_t$; broadcast $w_t$ to $S_t$
4:     Each $i \in S_t$ returns $\Delta_{t,i} = \theta_{t,i} - w_t$
5:     Update $w_{t+1} \leftarrow w_t + \sum_{i \in S_t} p_{t,i} \Delta_{t,i}$     (Eq. (2))
6:     Form $\Delta_t = [\sqrt{p_{t,i}} \Delta_{t,i}]_{i \in S_t}$; compute thin SVD $\Delta_t = U\Sigma V^\top$; set $U_t \leftarrow U_{(:,1:r)}$
7:     **for** each $i \in S_t$ **do**
8:         $r_{t,i} \leftarrow (I - U_t U_t^\top) \Delta_{t,i}$     (Eq. (6))
9:         $\tilde{v}_{t,i} \leftarrow V_i V_i^\top r_{t,i}$     (Eq. (7))
10:     **end for**
11:     $\bar{v}_t \leftarrow \sum_{j \in S_t} p_{t,j} \tilde{v}_{t,j}$
12:     **for** each $i \in S_t$ **do**
13:         Compute $\mathcal{E}_{t,i}$ by Eq. (8) and Eqs. (9)–(11); set $s_{t,i} \leftarrow \frac{1}{1+\lambda \mathcal{E}_{t,i}}$
14:         $z_{t+1,i} \leftarrow z_{t,i} + p_{t,i} s_{t,i} V_i^\top r_{t,i}$     (Eq. (12))
15:     **end for**
16:     For $i \notin S_t$, set $z_{t+1,i} \leftarrow z_{t,i}$
17: **end for**
18: Output trained model $w_T$ and influence bank $\{z_{T,i}\}_{i=1}^N$

---

**Algorithm 2** IDFT Unlearning for Client $u$

1: **Input:** trained $w_T$, bank $\{z_{T,i}\}$, delete client $u$, repair rounds $R$
2: Compute stored influence $\phi_{T,u} \leftarrow V_u z_{T,u}$; set $w^{(-u,0)} \leftarrow w_T - \phi_{T,u}$     (Eq. (13))
3: Run $R$ rounds of proximal federated optimization on clients $[N] \setminus \{u\}$ for objective in Eq. (14)
4: **Output:** final unlearned model $w^{-u}$

---

objective

$$
\begin{aligned}
F_t^{\backslash u}(w) &\triangleq \sum_{i \in S_t \backslash \{u\}} p_{t,i}^{\backslash u} f_i(w), \\
p_{t,i}^{\backslash u} &\triangleq \frac{n_i}{\sum_{j \in S_t \backslash \{u\}} n_j} \quad (i \in S_t \backslash \{u\}).
\end{aligned}
\quad (15)
$$

We assume each $f_i$ is twice differentiable and $(\mu, L)$-regular:

**Assumption 4.1** $((\mu, L)$-regularity). *For all $i \in [N]$ and all $w \in \mathbb{R}^d$, the Hessian satisfies $\mu I \preceq \nabla^2 f_i(w) \preceq LI$ for some constants $0 < \mu \leq L$.*

**Retained update map.** Under FedSGD with step size $\eta_t \in (0, 2/(L+\mu)]$, the leave-$u$-out retraining iterates follow

$$w_{t+1}^{\backslash u} = \mathcal{G}_t\left(w_t^{\backslash u}\right), \qquad \mathcal{G}_t(w) \triangleq w - \eta_t \nabla F_t^{\backslash u}(w). \quad (16)$$

**Residual influence to be removed.** IDFT logs the orthogonal residual $r_{t,u}$ of client $u$'s round-$t$ update (Sec. 3.2), and stores the shrunken, client-separable component $\hat{r}_{t,u} \triangleq$

$s_{t,u} V_u V_u^\top r_{t,u}$ with $s_{t,u} = 1/(1 + \lambda \mathcal{E}_{t,u})$. Define the *unremoved* residual in round $t$ as

$$\Delta_{t,u}^{\text{miss}} \triangleq p_{t,u}\big(r_{t,u} - \widehat{r}_{t,u}\big) = p_{t,u}\big(I - s_{t,u} V_u V_u^\top\big) r_{t,u}. \tag{17}$$

**A contractive surrogate for "IDFT-with-deletion".** We analyze the following *idealized* deletion-time trajectory, which applies the retained map $\mathcal{G}_t$ and injects precisely the residual that IDFT fails to remove:

$$\widetilde{w}_{t+1} = \mathcal{G}_t(\widetilde{w}_t) + \Delta_{t,u}^{\text{miss}}, \qquad \widetilde{w}_0 = w_0, \tag{18}$$

where $\Delta_{t,u}^{\text{miss}} = 0$ if $u \notin S_t$. This recursion formalizes the role of IDFT: the retrain gap is driven by the (structured) missed residuals, while the retained learning dynamics remain unchanged.

**Theorem 4.2** (Unlearning fidelity bound for IDFT)**.** *Let Assumption 4.1 hold and choose $\eta_t \in (0, 2/(L+\mu)]$ for all $t$. Define the contraction factor $q_t \triangleq \max_{\lambda \in [\mu, L]} |1 - \eta_t \lambda| \in (0,1)$. Let $w_T^{\setminus u}$ be the leave-$u$-out retraining iterate from* (16), *and let $\widetilde{w}_T$ be the IDFT deletion surrogate from* (18). *Then*

$$\big\|\widetilde{w}_T - w_T^{\setminus u}\big\|_2 \leq \sum_{t:\, u \in S_t} \Big(\prod_{k=t+1}^{T-1} q_k\Big) p_{t,u}$$

$$\times \Big(\big\|(I - V_u V_u^\top) r_{t,u}\big\|_2 + \frac{\lambda \mathcal{E}_{t,u}}{1 + \lambda \mathcal{E}_{t,u}} \|r_{t,u}\|_2\Big). \tag{19}$$

*Moreover, if the constrained repair runs $R$ steps of gradient descent on $\Psi(w) \triangleq F_{-u}(w) + \frac{\beta}{2}\|w - \widetilde{w}_T\|_2^2$ with step size $\alpha \in (0, 2/(L + \beta + \mu + \beta)]$, and outputs $w^{-u}$, then with $q_\beta \triangleq \max_{\lambda \in [\mu+\beta,\, L+\beta]} |1 - \alpha \lambda| \in (0,1)$,*

$$\|w^{-u} - w_\star^{\setminus u}\|_2 \leq \Big(\frac{\mu + 2\beta}{\mu + \beta} q_\beta^R + \frac{\beta}{\mu + \beta}\Big) \|\widetilde{w}_T - w_\star^{\setminus u}\|_2, \tag{20}$$

*where $w_\star^{\setminus u} \triangleq \arg\min_w F_{-u}(w)$ is the oracle retraining solution.*

*Proof sketch.* Lemma A.1 shows that each retained map $\mathcal{G}_t$ is a contraction with factor $q_t$ under Assumption 4.1. Lemma A.2 then yields that the deviation between the perturbed recursion (18) and the unperturbed retraining recursion (16) is bounded by a geometrically weighted sum of per-round perturbations $\|\Delta_{t,u}^{\text{miss}}\|_2$. Lemma A.3 upper bounds $\|\Delta_{t,u}^{\text{miss}}\|_2$ by the sum of (i) the influence-subspace approximation error $\|(I - V_u V_u^\top) r_{t,u}\|_2$ and (ii) the entanglement-induced shrinkage loss $(1 - s_{t,u})\|r_{t,u}\|_2$, where $1 - s_{t,u} = \lambda \mathcal{E}_{t,u}/(1 + \lambda \mathcal{E}_{t,u})$. Combining these three steps proves (19). Finally, Proposition A.4 gives the repair factor (20) by decomposing the error into optimization contraction towards the proximal minimizer and the explicit bias between the proximal minimizer and $w_\star^{\setminus u}$ under strong convexity. $\square$

## 4.2. Recoverability and Sample Complexity of the Shared Subspace

This section characterizes when the server-side shared subspace estimator $U_t$ (top-$r$ left singular vectors of the weighted update matrix $\Delta_t$) accurately recovers the underlying *population* shared subspace. The result quantifies an explicit *effective sample size* induced by the round-normalized weights $\{p_{t,i}\}_{i \in S_t}$ and yields a sharp leakage control for the decomposition step in Eq. (6).

**Setup and population subspace.** Fix a round $t$ with participating set $S_t$ of size $m \triangleq |S_t|$. Let $\{\Delta_{t,i} \in \mathbb{R}^d\}_{i \in S_t}$ be the collected client updates and $p_{t,i} = \frac{n_i}{\sum_{j \in S_t} n_j}$ the round-normalized weights. Define the weighted update matrix (as in Sec. 3.2)

$$\Delta_t \triangleq \big[\sqrt{p_{t,i}} \Delta_{t,i}\big]_{i \in S_t} \in \mathbb{R}^{d \times m}. \tag{21}$$

The left singular vectors of $\Delta_t$ coincide with the eigenvectors of the weighted empirical second moment

$$\widehat{C}_t \triangleq \Delta_t \Delta_t^\top = \sum_{i \in S_t} p_{t,i} \Delta_{t,i} \Delta_{t,i}^\top \in \mathbb{R}^{d \times d}. \tag{22}$$

We define the *population* second moment (conditional on the server state at round $t$) as

$$C_t \triangleq \mathbb{E}\big[\widehat{C}_t\big] = \sum_{i \in S_t} p_{t,i} \mathbb{E}\big[\Delta_{t,i} \Delta_{t,i}^\top\big]. \tag{23}$$

Let the spectral decomposition of $C_t$ be $C_t = U_{\star,t} \Lambda_{\star,t} U_{\star,t}^\top + U_{\star,t}^\perp \Lambda_{\perp,t} U_{\star,t}^{\perp\top}$, where $U_{\star,t} \in \mathbb{R}^{d \times r}$ contains the top-$r$ eigenvectors (orthonormal columns). The algorithm sets $U_t$ to the top-$r$ eigenvectors of $\widehat{C}_t$ (equivalently, the top-$r$ left singular vectors of $\Delta_t$).

We measure subspace error via the largest principal angle:

$$\sin \Theta_t \triangleq \big\|(I - U_t U_t^\top) U_{\star,t}\big\|_{\text{op}} = \big\|U_{\star,t}^{\perp\top} U_t\big\|_{\text{op}} \in [0,1]. \tag{24}$$

We also define the *effective sample size* induced by weights:

$$n_{\text{eff}}(t) \triangleq \frac{1}{\sum_{i \in S_t} p_{t,i}^2}, \qquad p_{t,\max} \triangleq \max_{i \in S_t} p_{t,i}. \tag{25}$$

**Assumption 4.3** (Conditional independence and bounded updates)**.** *Conditional on the server state at round $t$, the random vectors $\{\Delta_{t,i}\}_{i \in S_t}$ are independent and satisfy*

$$\|\Delta_{t,i}\|_2 \leq B_t \quad \text{almost surely for all } i \in S_t, \tag{26}$$

*for some finite constant $B_t > 0$.*

**Assumption 4.4** (Spectral gap of the population moment)**.** *Let $\lambda_{t,1} \geq \cdots \geq \lambda_{t,d}$ be the eigenvalues of $C_t$. The $r$-th gap is strictly positive:*

$$\gamma_t \triangleq \lambda_{t,r} - \lambda_{t,r+1} > 0. \tag{27}$$

### 4.2.1. MAIN RESULT: SUBSPACE RECOVERABILITY AND SAMPLE COMPLEXITY

**Theorem 4.5** (Recoverability of the shared subspace $U_t$). *Fix round $t$ and let Assumptions 4.3–4.4 hold. Let $U_t$ be the top-$r$ eigenvectors of $\widehat{C}_t$ in Eq. (22), and $U_{\star,t}$ the top-$r$ eigenvectors of $C_t$ in Eq. (23). Then, for any $\delta \in (0,1)$, with probability at least $1 - \delta$,*

$$\left\| \widehat{C}_t - C_t \right\|_{\mathrm{op}} \leq 2B_t^2 \sqrt{2 \log\left(\frac{2d}{\delta}\right) \sum_{i \in S_t} p_{t,i}^2} \tag{28}$$
$$+ \frac{4}{3} B_t^2 \, p_{t,\max} \, \log\left(\frac{2d}{\delta}\right).$$

*Moreover, under the same event,*

$$\sin \Theta_t \leq \min\left\{ 1, \ \frac{2}{\gamma_t} \| \widehat{C}_t - C_t \|_{\mathrm{op}} \right\}. \tag{29}$$

**Corollary 4.6** (Sample complexity in terms of $n_{\mathrm{eff}}(t)$). *Under the conditions of Theorem 4.5, if the weights are such that*

$$n_{\mathrm{eff}}(t) \geq \frac{128 B_t^4}{\gamma_t^2 \varepsilon^2} \log\left(\frac{2d}{\delta}\right) \quad and$$
$$p_{t,\max} \leq \frac{3\gamma_t \varepsilon}{16 B_t^2 \log\left(\frac{2d}{\delta}\right)}, \tag{30}$$

*then with probability at least $1 - \delta$ we have $\sin \Theta_t \leq \varepsilon$.*

**Corollary 4.7** (Leakage control for the decomposition step). *Under the event in Theorem 4.5, for any vector $x \in \mathrm{span}(U_{\star,t})$,*

$$\left\| (I - U_t U_t^\top) x \right\|_2 \leq \sin \Theta_t \, \|x\|_2. \tag{31}$$

*In particular, for any client $i \in S_t$, letting $x = U_{\star,t} U_{\star,t}^\top \Delta_{t,i}$ (the population-shared component of $\Delta_{t,i}$), the portion of shared signal that leaks into the residual $r_{t,i} = (I - U_t U_t^\top)\Delta_{t,i}$ is bounded by $\sin \Theta_t \| U_{\star,t} U_{\star,t}^\top \Delta_{t,i} \|_2$.*

*Proof sketch.* Equation (22) shows $\widehat{C}_t$ is a weighted sum of rank-one PSD matrices. Using Assumption 4.3, we bound each centered summand $X_{t,i} \triangleq p_{t,i}(\Delta_{t,i}\Delta_{t,i}^\top - \mathbb{E}[\Delta_{t,i}\Delta_{t,i}^\top])$ in operator norm and compute a tight variance proxy scaling with $\sum_i p_{t,i}^2$. Applying the (self-adjoint) matrix Bernstein inequality yields the concentration bound (28). Given $\|\widehat{C}_t - C_t\|_{\mathrm{op}}$, the sin-$\Theta$ perturbation bound (29) follows from a Davis–Kahan-type argument that compares the top-$r$ invariant subspaces of $C_t$ and $\widehat{C}_t$ via the eigengap $\gamma_t$. Finally, (31) is a direct projection inequality: $\|(I - U_t U_t^\top)U_{\star,t}\|_{\mathrm{op}} = \sin \Theta_t$ implies $\|(I - U_t U_t^\top)x\| \leq \sin \Theta_t \|x\|$ for all $x \in \mathrm{span}(U_{\star,t})$. Complete proofs are in Appendix A.2. $\square$

## 5. Experiments

### 5.1. Experimental Setup

**Datasets.** We evaluate IDFT on widely adopted federated-unlearning image benchmarks, including CIFAR-10, CIFAR-100, Fashion-MNIST, Caltech-101, and EuroSAT. We simulate cross-device FL with both IID and label-skew non-IID client partitions via Dirichlet($\alpha$), where smaller $\alpha$ indicates higher heterogeneity. Our main setting is client-level unlearning: after the global model converges, one client $u$ requests deletion, and we compare the unlearned model against retraining-from-scratch without $u$. Additional class- and sample-level protocols (for aligning with class-centric baselines) are described in Appendix B.1.

**Evaluation metrics.** Following standard federated-unlearning practice, we report: (i) *utility preservation* as accuracy on retained data (a.k.a. Retain/RA), (ii) *forgetting efficacy* as accuracy on deleted data (a.k.a. Forget/FA) and membership-inference attack success/accuracy (MIA), (iii) *retrain fidelity* via (a) parameter gap $\|w^{-u} - w^{\setminus u}\|_2$ and (b) accuracy deltas to the retrained model, and (iv) *efficiency* via unlearning wall-clock time, communication volume, and server-side extra storage. This aligns with the metric suites used in NoT/FUCRT/FUSED-style evaluations.

**Compared methods.** We compare against baselines covering distinct unlearning paradigms: class-aware representation transformation FUCRT (Guo et al., 2025), weight negation NoT (Khalil et al., 2025), and reversible knowledge overwriting with sparse adapters FUSED (Zhong et al., 2025). We additionally include commonly reported FU baselines appearing in recent comparison tables: FedEraser (Liu et al., 2021), ClientErase / projected gradient methods (Halimi et al., 2022), knowledge-distillation unlearning (FUKD) (Wu et al., 2022a), rapid retraining (Liu et al., 2022), sequential informed FU (SIFU) (Fraboni et al., 2024), class-discriminative pruning (Wang et al., 2022), VeriFi (Gao et al., 2024), auxiliary-module unlearning during learning (FedAU) (Gu et al., 2024), orthogonal-steepest-descent unlearning (FedOSD) (Pan et al., 2025), momentum degradation (MoDe) (Zhao et al., 2023), federated client unlearning at the feature level (FCU) (Deng et al., 2024), exact federated unlearning (Exact-Fun) (Xiong et al., 2023), and EraseClient (Wu et al., 2022b).

**Implementation details.** IDFT adds training-time logging and performs deletion by subtraction followed by $R \ll T$ proximal repair rounds. Default hyperparameters and tuning ranges for $(r, s, \lambda, a, b)$ and repair $(\beta, R)$ are provided in Appendix B.3.

### 5.2. Main Results

**Client-wise retrain fidelity.** As shown in Table 1(A), *IDFT achieves the lowest Avg. Gap on all five dataset/architecture*

*Table 1.* **Main results under two representative FU benchmarks.** (A) *Avg. Gap* ↓ follows the benchmark protocol and baseline results reported by NoT (Khalil et al., 2025). (B) Client-unlearning metrics follow the reversible FU benchmark and baseline results reported by FUSED (Zhong et al., 2025). IDFT results are evaluated under the same metric definitions.

**(A) Client-wise unlearning fidelity (IID, 10 clients): Avg. Gap ↓**

| Method | CIFAR-10 (CNN) | CIFAR-100 (CNN) | CIFAR-10 (ResNet-18) | CIFAR-100 (ResNet-18) | Caltech-101 (ViT-B/16) |
|---|---|---|---|---|---|
| FT | 1.01 | 1.35 | 4.72 | 10.81 | 17.76 |
| FedEraser (Liu et al., 2021) | 1.58 | 2.56 | n/a | n/a | n/a |
| FUKD (Wu et al., 2022a) | 4.09 | 8.21 | n/a | n/a | n/a |
| PGD (Halimi et al., 2022) | 0.98 | 1.28 | 4.68 | 5.83 | 13.78 |
| MoDE (Zhao et al., 2023) | 0.90 | 1.07 | 1.83 | 5.16 | 1.60 |
| FCU (Deng et al., 2024) | 0.81 | 1.47 | 0.84 | 2.51 | 1.61 |
| NoT (Khalil et al., 2025) | 0.29 | 0.75 | 1.78 | 4.73 | 0.79 |
| **IDFT (ours)** | 0.26 | 0.72 | 0.78 | 2.33 | 0.68 |

**(B) Reversible FU benchmark (client unlearning; label-flipping attacks)**

| Dataset | Method | RA↑ | FA↓ | ReA↓ | MIA↓ | Comm↓ | Comp↓ |
|---|---|---|---|---|---|---|---|
| FashionMNIST-LeNet | Retrain | 0.995 | 0.00 | 0.77 | 0.85 | 177K | 210.15 |
| | FedEraser (Liu et al., 2021) | 0.996 | 0.00 | 0.94 | 0.47 | 177K | 178.66 |
| | Exact-Fun (Xiong et al., 2023) | 0.994 | 0.00 | 0.96 | 0.70 | 177K | 298.28 |
| | EraseClient (Wu et al., 2022b) | 0.990 | 0.11 | 0.96 | 0.70 | 177K | 26.31 |
| | FUSED (Zhong et al., 2025) | 0.992 | 0.00 | 0.97 | 0.68 | 11K | 158.96 |
| | **IDFT (ours)** | 0.997 | 0.00 | 0.94 | 0.62 | 13K | 25.80 |
| CIFAR10-ResNet18 | Retrain | 0.71 | 0.04 | 0.49 | 0.78 | 42.73M | 434.39 |
| | FedEraser (Liu et al., 2021) | 0.67 | 0.04 | 0.48 | 0.67 | 42.73M | 990.91 |
| | Exact-Fun (Xiong et al., 2023) | 0.65 | 0.05 | 0.41 | 0.43 | 42.73M | 1211.74 |
| | EraseClient (Wu et al., 2022b) | 0.64 | 0.06 | 0.56 | 0.78 | 42.73M | 233.45 |
| | FUSED (Zhong et al., 2025) | 0.67 | 0.05 | 0.42 | 0.65 | 0.98M | 262.20 |
| | **IDFT (ours)** | 0.68 | 0.04 | 0.40 | 0.45 | 1.05M | 225.10 |
| CIFAR100-ResNet18 | Retrain | 0.39 | 0.01 | 0.18 | 0.22 | 42.91M | 443.86 |
| | FedEraser (Liu et al., 2021) | 0.19 | 0.01 | 0.13 | 0.28 | 42.91M | 1000.75 |
| | Exact-Fun (Xiong et al., 2023) | 0.25 | 0.00 | 0.17 | 0.08 | 42.91M | 1598.55 |
| | EraseClient (Wu et al., 2022b) | 0.35 | 0.01 | 0.20 | 0.36 | 42.91M | 235.65 |
| | FUSED (Zhong et al., 2025) | 0.36 | 0.01 | 0.14 | 0.48 | 0.98M | 276.59 |
| | **IDFT (ours)** | 0.38 | 0.00 | 0.12 | 0.10 | 1.10M | 230.40 |

*pairs*, indicating consistently tighter matching to the leave-$u$-out retraining solution. Compared to the strongest prior baseline per column, IDFT reduces the gap by a ∼4–14% margin (e.g., $0.29 \to 0.26$ on CIFAR-10 CNN, $0.84 \to 0.78$ on CIFAR-10 ResNet-18, and $0.79 \to 0.68$ on Caltech-101 ViT), supporting that training-time influence logging meaningfully shrinks deletion-time residuals.

**Utility–forgetting–efficiency trade-off.** Table 1(B) shows that IDFT preserves utility (RA comparable to retrain/Fed-Eraser) while maintaining strong forgetting (FA at the best level on CIFAR10/100 and tied-best on FashionMNIST). Crucially, IDFT delivers the best ReA and lowest computation on CIFAR10/100, and substantially reduces communication relative to checkpoint/history-heavy methods (∼40×

smaller than FedEraser on CIFAR10/100), while remaining close to adapter-based FUSED in communication. Overall, IDFT provides a favorable fidelity–efficiency frontier by combining subtraction-based removal with a short repair.

### 5.3. Ablation and Analysis

**Single-factor ablation.** We perform one-factor ablations of IDFT to verify the contribution of each design choice. Table 2 shows **all components contribute complementary gains**. Removing subspace routing ($r=0$) sharply reduces fidelity (Avg. Gap $+0.27$) and worsens privacy (ReA/MIA increase), confirming the need to filter shared cross-client information. Reducing bank dimension ($s=32$) causes consistent moderate degradation, showing *bank capacity* matters

*Table 2.* **Single-factor ablation on CIFAR10-ResNet18.** Avg. Gap is from the IID 10-client fidelity benchmark; RA/FA/ReA/MIA are from the label-flipping benchmark.

| Variant | Avg. Gap ↓ | RA↑ | FA↓ | ReA↓ | MIA↓ |
|---|---|---|---|---|---|
| **IDFT (full)** | 0.78 | 0.68 | 0.04 | 0.40 | 0.45 |
| w/o shared subspace (set $r{=}0$) | 1.05 (+0.27) | 0.66 (-0.02) | 0.04 (+0.00) | 0.44 (+0.04) | 0.50 (+0.05) |
| smaller influence bank ($s{=}32$) | 0.86 (+0.08) | 0.67 (-0.01) | 0.04 (+0.00) | 0.41 (+0.01) | 0.46 (+0.01) |
| w/o entanglement shrinkage (set $\lambda{=}0$) | 0.84 (+0.06) | 0.66 (-0.02) | 0.04 (+0.00) | 0.42 (+0.02) | 0.47 (+0.02) |
| w/o covariance term (set $a{=}0$) | 0.89 (+0.11) | 0.67 (-0.01) | 0.05 (+0.01) | 0.43 (+0.03) | 0.49 (+0.04) |
| w/o co-adaptation term (set $b{=}0$) | 0.88 (+0.10) | 0.67 (-0.01) | 0.04 (+0.00) | 0.43 (+0.03) | 0.48 (+0.03) |
| w/o leakage term (ignore projection error) | 0.90 (+0.12) | 0.66 (-0.02) | 0.04 (+0.00) | 0.44 (+0.04) | 0.50 (+0.05) |
| w/o repair (set $R{=}0$) | 1.22 (+0.44) | 0.64 (-0.04) | 0.05 (+0.01) | 0.47 (+0.07) | 0.56 (+0.11) |

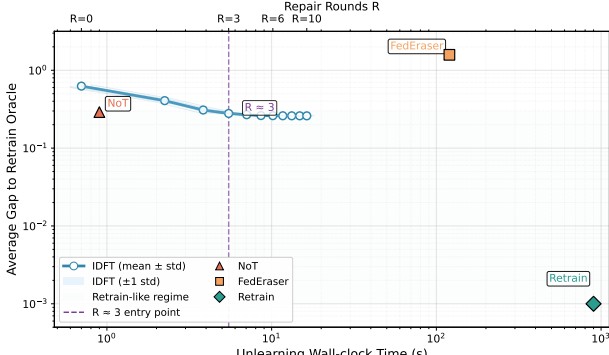

*Figure 1.* **Gap–cost trajectory under client deletion.** IDFT (solid curve) shows mean ± std across runs; faint curves are individual runs. Baselines are single points (unlearning time, Avg. Gap). The shaded region indicates a *retrain-like* regime.

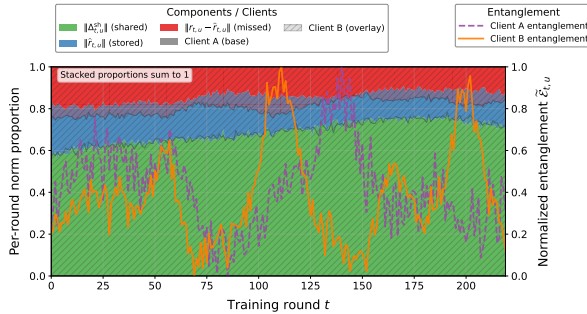

*Figure 2.* **Stacked area visualization of removability over rounds.** Each stack shows the per-round proportion of (shared) $\rho^{\mathrm{sh}}_{t,u}$, (stored removable) $\rho^{\mathrm{st}}_{t,u}$, and (missed) $\rho^{\mathrm{miss}}_{t,u}$ for two representative clients. Dashed lines show normalized entanglement $\widetilde{\mathcal{E}}_{t,u}$.

for capturing client-specific influence. Disabling entanglement shrinkage worsens ReA/MIA and increases retrain gaps, confirming its role in suppressing non-removable components. **Repair is crucial**: removing it ($R{=}0$) yields the largest overall degradation, showing subtraction alone is insufficient, while constrained repair closes the gap.

**Gap–Cost Trajectory of Unlearning: Fast Remove with Short Repair.** After deleting a client, does the retrain fidelity gap drop rapidly with only a few repair rounds, and is the unlearning cost substantially lower than retraining or history-heavy baselines? We delete one client after training convergence and run IDFT with repair rounds $R \in \{0, 1, \ldots, 10\}$. For each $R$, we record (i) *unlearning wall-clock time* from request to output, and (ii) *Avg. Gap* (lower is better) against the leave-one-out retraining oracle.

Fig. 1 shows that IDFT drives Avg. Gap into the retrain-like regime within $R = 3\text{–}5$ repair rounds, with diminishing gains thereafter. It matches or improves fidelity at orders-of-magnitude lower wall-clock time than history-heavy methods (e.g., FedEraser) or full retraining, enabled by *single-shot subtraction* plus a *short anchored repair*. In contrast, fast one-shot post-hoc baselines typically plateau at a larger gap, suggesting residual entanglement requires iterative correction beyond subtraction alone.

**Visualizing Removability.** IDFT posits that the shared subspace $U_t$ absorbs cross-client covarying directions, leaving a client-separable residual that is amenable to subtraction. To test this claim, we track (for one low-entanglement and one high-entanglement client) the *per-round* allocation of update magnitude into three components: the shared part $\|\Delta^{\mathrm{sh}}_{t,u}\|_2$, the stored removable part $\|\widehat{r}_{t,u}\|_2$, and the missed residual $\|r_{t,u} - \widehat{r}_{t,u}\|_2$, plotted as stacked proportions over rounds with an overlay of normalized entanglement $\widetilde{\mathcal{E}}_{t,u}$ (see Appendix B.4 for full definitions). Fig. 2 shows that the shared component consistently dominates, the stored removable component remains stable and non-trivial (supporting subtraction), and the missed portion stays small but spikes in sync with entanglement—consistent with $\Delta^{\mathrm{miss}}_{t,u}$ capturing harder-to-unlearn residuals that motivate short repair.

# 6. Conclusion

We tackle client-level federated unlearning under entangled client influence from iterative aggregation. IDFT logs client-separable influence during training, enabling subtraction plus short anchored repair with theory tying retrain fidelity to missed residuals. Experiments show a better fidelity–cost trade-off than strong baselines. Future work will extend this view to personalization and verifiable/enforceable unlearning in production FL.

## Impact Statement

This paper advances federated unlearning, a capability relevant to privacy regulation, data-governance compliance, and user deletion requests in decentralized learning systems. While IDFT aims to reduce the cost of removing client influence, approximate unlearning should not be interpreted as a substitute for formal legal compliance or independent auditing. We encourage future deployments to pair unlearning mechanisms with verification, access control, and transparent reporting of residual privacy risk.

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

# A. Proofs

## A.1. Full Proofs for Theorem 4.2

**Preliminaries.** We use the standard mean-value representation for gradients under Hessian bounds. Throughout, $\|\cdot\|_2$ is the Euclidean norm and $\|\cdot\|_{\mathrm{op}}$ is the spectral norm. For a twice differentiable function $g : \mathbb{R}^d \to \mathbb{R}$, define its gradient step map $\mathcal{T}_\eta(w) \triangleq w - \eta \nabla g(w)$.

### A.1.1. LEMMA: CONTRACTION OF A GRADIENT STEP UNDER $(\mu, L)$-REGULARITY

**Lemma A.1** (Gradient-step contraction). *Let $g$ be twice differentiable and satisfy $\mu I \preceq \nabla^2 g(w) \preceq LI$ for all $w$. Fix any step size $\eta > 0$ and define $q(\eta) \triangleq \max_{\lambda \in [\mu, L]} |1 - \eta\lambda|$. Then for all $x, y \in \mathbb{R}^d$,*

$$\|\mathcal{T}_\eta(x) - \mathcal{T}_\eta(y)\|_2 \ \leq \ q(\eta)\, \|x - y\|_2. \tag{32}$$

*In particular, if $\eta \in (0, 2/(L + \mu)]$, then $q(\eta) \in (0, 1)$.*

*Proof.* Fix $x, y$. Consider the segment $x(\tau) = y + \tau(x - y)$ for $\tau \in [0, 1]$. By the fundamental theorem of calculus applied

to the vector field $\nabla g$,

$$\nabla g(x) - \nabla g(y) = \int_0^1 \nabla^2 g\big(x(\tau)\big)\,(x-y)\,d\tau \;\triangleq\; H_{x,y}\,(x-y), \tag{33}$$

where $H_{x,y} \triangleq \int_0^1 \nabla^2 g(x(\tau))\,d\tau$ is symmetric and satisfies $\mu I \preceq H_{x,y} \preceq LI$ by convexity of the PSD cone and the assumed bounds. Therefore,

$$\mathcal{T}_\eta(x) - \mathcal{T}_\eta(y) = (x-y) - \eta(\nabla g(x) - \nabla g(y)) = (I - \eta H_{x,y})(x-y). \tag{34}$$

Taking norms and using $\|I - \eta H_{x,y}\|_{\mathrm{op}} = \max_{\lambda \in \sigma(H_{x,y})} |1 - \eta\lambda|$ with $\sigma(H_{x,y}) \subset [\mu, L]$ yields

$$\begin{aligned}
\|\mathcal{T}_\eta(x) - \mathcal{T}_\eta(y)\|_2 &\le \|I - \eta H_{x,y}\|_{\mathrm{op}}\|x-y\|_2 \\
&\le \max_{\lambda \in [\mu, L]} |1 - \eta\lambda|\,\|x-y\|_2 \\
&= q(\eta)\|x-y\|_2.
\end{aligned} \tag{35}$$

If $\eta \in (0, 2/(L+\mu)]$, then $|1 - \eta\mu| \le 1 - \eta\mu < 1$ and $|1 - \eta L| \le 1 - \eta\mu < 1$, hence $q(\eta) < 1$. $\qquad\square$

### A.1.2. LEMMA: ACCUMULATION OF ADDITIVE PERTURBATIONS IN A CONTRACTIVE RECURSION

**Lemma A.2** (Perturbation accumulation). *Let $\{\mathcal{G}_t\}_{t=0}^{T-1}$ be maps on $\mathbb{R}^d$ such that $\|\mathcal{G}_t(x) - \mathcal{G}_t(y)\|_2 \le q_t\|x-y\|_2$ for all $x, y$, with $q_t \in [0,1)$. Consider two sequences:*

$$x_{t+1} = \mathcal{G}_t(x_t), \quad x_0 \in \mathbb{R}^d, \qquad y_{t+1} = \mathcal{G}_t(y_t) + \delta_t, \quad y_0 = x_0, \tag{36}$$

*where $\delta_t \in \mathbb{R}^d$ are arbitrary perturbations. Then*

$$\|y_T - x_T\|_2 \le \sum_{t=0}^{T-1} \Big(\prod_{k=t+1}^{T-1} q_k\Big) \|\delta_t\|_2. \tag{37}$$

*Proof.* Define $e_t \triangleq y_t - x_t$. Since $e_0 = 0$, we have

$$e_{t+1} = y_{t+1} - x_{t+1} = \mathcal{G}_t(y_t) + \delta_t - \mathcal{G}_t(x_t). \tag{38}$$

Taking norms and using contractivity,

$$\|e_{t+1}\|_2 \le \|\mathcal{G}_t(y_t) - \mathcal{G}_t(x_t)\|_2 + \|\delta_t\|_2 \le q_t\|e_t\|_2 + \|\delta_t\|_2. \tag{39}$$

Unrolling (39) yields

$$\|e_T\|_2 \le \|\delta_{T-1}\|_2 + q_{T-1}\|\delta_{T-2}\|_2 + q_{T-1}q_{T-2}\|\delta_{T-3}\|_2 + \cdots + \Big(\prod_{k=1}^{T-1} q_k\Big)\|\delta_0\|_2 \tag{40}$$

$$= \sum_{t=0}^{T-1}\Big(\prod_{k=t+1}^{T-1} q_k\Big)\|\delta_t\|_2, \tag{41}$$

which proves the claim. $\qquad\square$

### A.1.3. LEMMA: DECOMPOSITION OF THE MISSED RESIDUAL UNDER SHRINKAGE AND SUBSPACE PROJECTION

**Lemma A.3** (Missed-residual decomposition). *Let $V \in \mathbb{R}^{d \times s}$ have orthonormal columns ($V^\top V = I_s$), let $s \in (0, 1]$, and let $r \in \mathbb{R}^d$. Then*

$$\|(I - sVV^\top)r\|_2 \le \|(I - VV^\top)r\|_2 + (1-s)\|r\|_2. \tag{42}$$

*Proof.* Decompose $r$ into orthogonal components: $r = VV^\top r + (I - VV^\top)r$ and $\langle VV^\top r, (I - VV^\top)r \rangle = 0$. Then

$$(I - sVV^\top)r = (I - VV^\top)r + (1 - s)VV^\top r. \tag{43}$$

Taking norms and applying the triangle inequality,

$$\begin{aligned}
\|(I - sVV^\top)r\|_2 &\leq \|(I - VV^\top)r\|_2 + (1 - s)\|VV^\top r\|_2 \\
&\leq \|(I - VV^\top)r\|_2 + (1 - s)\|r\|_2.
\end{aligned} \tag{44}$$

$\square$

### A.1.4. PROPOSITION: EXPLICIT BIAS OF THE PROXIMAL ANCHOR

**Proposition A.4** (Proximal bias towards the anchor). *Let $F_{-u}$ satisfy Assumption 4.1 (hence $F_{-u}$ is $\mu$-strongly convex). Fix an anchor $w_0 \in \mathbb{R}^d$ and $\beta > 0$, and define the proximal objective $\Psi(w) \triangleq F_{-u}(w) + \frac{\beta}{2}\|w - w_0\|_2^2$. Let $w_\star^{\setminus u} \triangleq \arg\min_w F_{-u}(w)$ and $w_\star^\beta \triangleq \arg\min_w \Psi(w)$. Then*

$$\|w_\star^\beta - w_\star^{\setminus u}\|_2 \leq \frac{\beta}{\mu + \beta}\|w_0 - w_\star^{\setminus u}\|_2. \tag{45}$$

*Moreover, if $w^{(R)}$ is obtained by $R$ steps of gradient descent on $\Psi$ with step size $\alpha \in (0, 2/(L + \beta + \mu + \beta)]$ from initialization $w^{(0)} = w_0$, then with $q_\beta = \max_{\lambda \in [\mu + \beta, L + \beta]} |1 - \alpha\lambda| \in (0, 1)$,*

$$\|w^{(R)} - w_\star^{\setminus u}\|_2 \leq \left(\frac{\mu + 2\beta}{\mu + \beta}q_\beta^R + \frac{\beta}{\mu + \beta}\right)\|w_0 - w_\star^{\setminus u}\|_2. \tag{46}$$

*Proof.* **Step 1 (bias bound).** The optimality conditions are $\nabla F_{-u}(w_\star^{\setminus u}) = 0$ and $\nabla F_{-u}(w_\star^\beta) + \beta(w_\star^\beta - w_0) = 0$. Subtracting gives

$$\nabla F_{-u}(w_\star^\beta) - \nabla F_{-u}(w_\star^{\setminus u}) = -\beta(w_\star^\beta - w_0). \tag{47}$$

Take inner product of both sides with $d \triangleq w_\star^\beta - w_\star^{\setminus u}$:

$$\langle d, \nabla F_{-u}(w_\star^\beta) - \nabla F_{-u}(w_\star^{\setminus u}) \rangle = -\beta\langle d, w_\star^\beta - w_0 \rangle = -\beta\|d\|_2^2 - \beta\langle d, w_\star^{\setminus u} - w_0 \rangle. \tag{48}$$

By $\mu$-strong convexity (equivalently, $\mu$-strong monotonicity of the gradient),

$$\langle d, \nabla F_{-u}(w_\star^\beta) - \nabla F_{-u}(w_\star^{\setminus u}) \rangle \geq \mu\|d\|_2^2. \tag{49}$$

Combining with (48) yields

$$\mu\|d\|_2^2 \leq -\beta\|d\|_2^2 + \beta\|d\|_2\|w_0 - w_\star^{\setminus u}\|_2, \tag{50}$$

hence $(\mu + \beta)\|d\|_2 \leq \beta\|w_0 - w_\star^{\setminus u}\|_2$, proving the first claim.

**Step 2 (optimization contraction).** By Assumption 4.1, $F_{-u}$ is $L$-smooth and $\mu$-strongly convex, so $\Psi$ is $(L + \beta)$-smooth and $(\mu + \beta)$-strongly convex. Applying Lemma A.1 to $\Psi$ gives

$$\|w^{(R)} - w_\star^\beta\|_2 \leq q_\beta^R\|w^{(0)} - w_\star^\beta\|_2. \tag{51}$$

Now,

$$\begin{aligned}
\|w^{(0)} - w_\star^\beta\|_2 &\leq \|w^{(0)} - w_\star^{\setminus u}\|_2 + \|w_\star^\beta - w_\star^{\setminus u}\|_2 \\
&\leq \left(1 + \frac{\beta}{\mu + \beta}\right)\|w_0 - w_\star^{\setminus u}\|_2 \\
&= \frac{\mu + 2\beta}{\mu + \beta}\|w_0 - w_\star^{\setminus u}\|_2.
\end{aligned} \tag{52}$$

Finally, by triangle inequality,

$$\begin{aligned}
\|w^{(R)} - w_\star^{\setminus u}\|_2 &\leq \|w^{(R)} - w_\star^\beta\|_2 + \|w_\star^\beta - w_\star^{\setminus u}\|_2 \\
&\leq q_\beta^R\frac{\mu + 2\beta}{\mu + \beta}\|w_0 - w_\star^{\setminus u}\|_2 + \frac{\beta}{\mu + \beta}\|w_0 - w_\star^{\setminus u}\|_2,
\end{aligned} \tag{53}$$

which proves the second claim. $\square$

A.1.5. PROOF OF THEOREM 4.2

*Proof of Theorem 4.2.* **Step 1 (apply contraction to retraining vs. missed-residual recursion).** We apply Lemma A.1 to each $\mathcal{G}_t$ in (16). Under Assumption 4.1, $F_t^{\setminus u}$ has Hessian in $[\mu, L]$ (as a convex combination of $\{f_i\}$), so Lemma A.1 yields that $\mathcal{G}_t$ is $q_t$-contractive:

$$\|\mathcal{G}_t(x) - \mathcal{G}_t(y)\|_2 \leq q_t \|x - y\|_2. \tag{54}$$

Now instantiate Lemma A.2 with $x_t = w_t^{\setminus u}$, $y_t = \widetilde{w}_t$, and perturbation $\delta_t = \Delta_{t,u}^{\mathrm{miss}}$:

$$\|\widetilde{w}_T - w_T^{\setminus u}\|_2 \leq \sum_{t=0}^{T-1} \Big( \prod_{k=t+1}^{T-1} q_k \Big) \|\Delta_{t,u}^{\mathrm{miss}}\|_2. \tag{55}$$

Since $\Delta_{t,u}^{\mathrm{miss}} = 0$ when $u \notin S_t$, the sum restricts to $t$ with $u \in S_t$.

**Step 2 (bound each missed residual by leakage + shrink loss).** For $t$ with $u \in S_t$, combine the definition (17) with Lemma A.3:

$$\begin{aligned}
\|\Delta_{t,u}^{\mathrm{miss}}\|_2 &= p_{t,u} \|(I - s_{t,u} V_u V_u^\top) r_{t,u}\|_2 \\
&\leq p_{t,u} \Big( \|(I - V_u V_u^\top) r_{t,u}\|_2 + (1 - s_{t,u}) \|r_{t,u}\|_2 \Big).
\end{aligned} \tag{56}$$

Using $s_{t,u} = 1/(1 + \lambda \mathcal{E}_{t,u})$ gives

$$1 - s_{t,u} = \frac{\lambda \mathcal{E}_{t,u}}{1 + \lambda \mathcal{E}_{t,u}}. \tag{57}$$

Substituting (56) into the accumulated bound completes the proof of (19).

**Step 3 (repair factor).** Equation (20) follows directly from Proposition A.4 by setting the anchor $w_0 = \widetilde{w}_T$ and identifying the repair iterate $w^{-u}$ with $w^{(R)}$ in the proposition. $\square$

## A.2. Full Proofs for Shared-Subspace Recoverability

A.2.1. A MATRIX BERNSTEIN INEQUALITY

We use the following standard self-adjoint matrix Bernstein inequality (Tropp, 2012).

**Lemma A.5** (Matrix Bernstein (self-adjoint)). *Let $\{X_k\}_{k=1}^m$ be independent, mean-zero, self-adjoint random matrices in $\mathbb{R}^{d \times d}$. Assume $\|X_k\|_{\mathrm{op}} \leq R$ almost surely for all $k$ and define the variance parameter*

$$v \triangleq \left\| \sum_{k=1}^m \mathbb{E}[X_k^2] \right\|_{\mathrm{op}}. \tag{58}$$

*Then for all $t \geq 0$,*

$$\mathbb{P}\left( \left\| \sum_{k=1}^m X_k \right\|_{\mathrm{op}} \geq t \right) \leq 2d \exp\left( \frac{-t^2}{2v + \frac{2}{3} R t} \right). \tag{59}$$

A.2.2. CONCENTRATION OF THE WEIGHTED EMPIRICAL SECOND MOMENT

**Lemma A.6** (Weighted covariance concentration). *Fix round $t$ and suppose Assumption 4.3 holds. Define $\widehat{C}_t$ and $C_t$ as in Eqs. (22)–(23). Then for any $\delta \in (0,1)$, with probability at least $1 - \delta$,*

$$\|\widehat{C}_t - C_t\|_{\mathrm{op}} \leq 2B_t^2 \sqrt{2 \log\Big(\frac{2d}{\delta}\Big) \sum_{i \in S_t} p_{t,i}^2} + \frac{4}{3} B_t^2 \, p_{t,\max} \log\Big(\frac{2d}{\delta}\Big). \tag{60}$$

*Proof.* For each $i \in S_t$, define the centered self-adjoint matrix

$$X_{t,i} \triangleq p_{t,i} \Big( \Delta_{t,i} \Delta_{t,i}^\top - \mathbb{E}[\Delta_{t,i} \Delta_{t,i}^\top] \Big). \tag{61}$$

Then $\mathbb{E}[X_{t,i}] = 0$ and

$$\widehat{C}_t - C_t = \sum_{i \in S_t} X_{t,i}. \tag{62}$$

**Step 1: almost-sure bound on $\|X_{t,i}\|_{\mathrm{op}}$.** Using the triangle inequality, submultiplicativity, and $\|vv^\top\|_{\mathrm{op}} = \|v\|_2^2$,

$$
\begin{aligned}
\|X_{t,i}\|_{\mathrm{op}} &\leq p_{t,i}\Big(\|\Delta_{t,i}\Delta_{t,i}^\top\|_{\mathrm{op}} + \big\|\mathbb{E}[\Delta_{t,i}\Delta_{t,i}^\top]\big\|_{\mathrm{op}}\Big) \\
&\leq p_{t,i}\Big(\|\Delta_{t,i}\|_2^2 + \mathbb{E}\|\Delta_{t,i}\|_2^2\Big).
\end{aligned}
\tag{63}
$$

By Assumption 4.3, $\|\Delta_{t,i}\|_2 \leq B_t$ almost surely, hence $\mathbb{E}\|\Delta_{t,i}\|_2^2 \leq B_t^2$ and

$$\|X_{t,i}\|_{\mathrm{op}} \leq 2p_{t,i}B_t^2 \leq 2p_{t,\max}B_t^2. \tag{64}$$

Thus Lemma A.5 applies with

$$R \triangleq 2p_{t,\max}B_t^2. \tag{65}$$

**Step 2: variance proxy $v$.** First note that for any self-adjoint $A$, $\|A^2\|_{\mathrm{op}} = \|A\|_{\mathrm{op}}^2$. Using (64), we have almost surely

$$
\begin{aligned}
\big\|\big(\Delta_{t,i}\Delta_{t,i}^\top - \mathbb{E}[\Delta_{t,i}\Delta_{t,i}^\top]\big)^2\big\|_{\mathrm{op}} &\leq \big\|\Delta_{t,i}\Delta_{t,i}^\top - \mathbb{E}[\Delta_{t,i}\Delta_{t,i}^\top]\big\|_{\mathrm{op}}^2 \\
&\leq \big(2B_t^2\big)^2 \\
&= 4B_t^4.
\end{aligned}
\tag{66}
$$

Multiplying by $p_{t,i}^2$ yields

$$\|X_{t,i}^2\|_{\mathrm{op}} = p_{t,i}^2\big\|\big(\Delta_{t,i}\Delta_{t,i}^\top - \mathbb{E}[\Delta_{t,i}\Delta_{t,i}^\top]\big)^2\big\|_{\mathrm{op}} \leq 4p_{t,i}^2B_t^4. \tag{67}$$

Taking expectations and summing,

$$
\begin{aligned}
\Big\|\sum_{i \in S_t}\mathbb{E}[X_{t,i}^2]\Big\|_{\mathrm{op}} &\leq \sum_{i \in S_t}\big\|\mathbb{E}[X_{t,i}^2]\big\|_{\mathrm{op}} \\
&\leq \sum_{i \in S_t}\mathbb{E}\|X_{t,i}^2\|_{\mathrm{op}} \\
&\leq 4B_t^4 \sum_{i \in S_t} p_{t,i}^2.
\end{aligned}
\tag{68}
$$

Thus Lemma A.5 applies with variance parameter

$$v \triangleq 4B_t^4 \sum_{i \in S_t} p_{t,i}^2. \tag{69}$$

**Step 3: tail inversion.** Apply Lemma A.5 to (62) with $R$ from (65) and $v$ from (69). Let $L_\delta \triangleq \log\big(\frac{2d}{\delta}\big)$. Choose

$$t_\delta \triangleq \sqrt{2vL_\delta} + \frac{2}{3}RL_\delta. \tag{70}$$

Then a standard calculation verifies the exponent satisfies $\frac{t_\delta^2}{2v + \frac{2}{3}Rt_\delta} \geq L_\delta$, hence $\mathbb{P}(\|\sum_i X_{t,i}\|_{\mathrm{op}} \geq t_\delta) \leq \delta$. Substituting (65)–(69) into (70) gives exactly (60). $\qquad\square$

### A.2.3. A DAVIS–KAHAN-TYPE SIN-$\Theta$ BOUND

**Lemma A.7** (Sin-$\Theta$ bound via eigengap)**.** *Let $A, \widehat{A} \in \mathbb{R}^{d \times d}$ be symmetric with $\widehat{A} = A + E$. Let $U \in \mathbb{R}^{d \times r}$ and $\widehat{U} \in \mathbb{R}^{d \times r}$ contain the top-$r$ eigenvectors of $A$ and $\widehat{A}$ (orthonormal columns). Let $\gamma \triangleq \lambda_r(A) - \lambda_{r+1}(A) > 0$. Then*

$$\big\|(I - \widehat{U}\widehat{U}^\top)U\big\|_{\mathrm{op}} \leq \min\Big\{1, \frac{2}{\gamma}\|E\|_{\mathrm{op}}\Big\}. \tag{71}$$

*Proof.* If $\|E\|_{\mathrm{op}} \geq \gamma/2$, then the right-hand side of (71) is at least 1 and the inequality holds trivially since the left-hand side is in $[0, 1]$ by definition. Assume henceforth

$$\|E\|_{\mathrm{op}} < \frac{\gamma}{2}. \tag{72}$$

Let $U_\perp \in \mathbb{R}^{d \times (d-r)}$ be an orthonormal complement of $U$, and write the eigen-decomposition

$$A = U\Lambda_1 U^\top + U_\perp \Lambda_2 U_\perp^\top, \tag{73}$$

where $\Lambda_1 = \mathrm{diag}(\lambda_1(A), \ldots, \lambda_r(A))$ and $\Lambda_2 = \mathrm{diag}(\lambda_{r+1}(A), \ldots, \lambda_d(A))$. Similarly, $\widehat{A}\widehat{U} = \widehat{U}\widehat{\Lambda}_1$ for some diagonal $\widehat{\Lambda}_1$. Premultiply $\widehat{A}\widehat{U} = \widehat{U}\widehat{\Lambda}_1$ by $U_\perp^\top$ and use $\widehat{A} = A + E$:

$$U_\perp^\top A\widehat{U} + U_\perp^\top E\widehat{U} = U_\perp^\top \widehat{U}\,\widehat{\Lambda}_1. \tag{74}$$

Using (73) and $U_\perp^\top U = 0$, we have $U_\perp^\top A = \Lambda_2 U_\perp^\top$, hence

$$\Lambda_2\,(U_\perp^\top \widehat{U}) + U_\perp^\top E\widehat{U} = (U_\perp^\top \widehat{U})\,\widehat{\Lambda}_1. \tag{75}$$

Rearranging,

$$(\Lambda_2 - \widehat{\Lambda}_1)\,(U_\perp^\top \widehat{U}) = -U_\perp^\top E\widehat{U}. \tag{76}$$

Taking operator norms and using $\|U_\perp^\top E\widehat{U}\|_{\mathrm{op}} \leq \|E\|_{\mathrm{op}}$,

$$\|U_\perp^\top \widehat{U}\|_{\mathrm{op}} \leq \|(\Lambda_2 - \widehat{\Lambda}_1)^\dagger\|_{\mathrm{op}}\,\|E\|_{\mathrm{op}}, \tag{77}$$

where $(\cdot)^\dagger$ denotes the Moore–Penrose pseudoinverse.

It remains to lower bound the singular values of $(\Lambda_2 - \widehat{\Lambda}_1)$. By Weyl's inequality for eigenvalues,

$$\left|\widehat{\lambda}_j(\widehat{A}) - \lambda_j(A)\right| \leq \|E\|_{\mathrm{op}} \qquad \text{for all } j \in [d]. \tag{78}$$

Therefore, for any $a \in \{r+1, \ldots, d\}$ and $b \in \{1, \ldots, r\}$,

$$\lambda_a(A) - \widehat{\lambda}_b(\widehat{A}) \leq \lambda_{r+1}(A) - \left(\lambda_r(A) - \|E\|_{\mathrm{op}}\right) = -(\gamma - \|E\|_{\mathrm{op}}), \tag{79}$$

and similarly

$$\widehat{\lambda}_b(\widehat{A}) - \lambda_a(A) \geq \left(\lambda_r(A) - \|E\|_{\mathrm{op}}\right) - \lambda_{r+1}(A) = \gamma - \|E\|_{\mathrm{op}}. \tag{80}$$

Thus every diagonal entry of $(\Lambda_2 - \widehat{\Lambda}_1)$ has magnitude at least $\gamma - \|E\|_{\mathrm{op}}$, implying

$$\|(\Lambda_2 - \widehat{\Lambda}_1)^\dagger\|_{\mathrm{op}} \leq \frac{1}{\gamma - \|E\|_{\mathrm{op}}} \leq \frac{2}{\gamma}, \tag{81}$$

where the last inequality uses (72). Combining (77) and (81) yields $\|U_\perp^\top \widehat{U}\|_{\mathrm{op}} \leq \frac{2}{\gamma}\|E\|_{\mathrm{op}}$. Finally, $\|(I - \widehat{U}\widehat{U}^\top)U\|_{\mathrm{op}} = \|U_\perp^\top \widehat{U}\|_{\mathrm{op}}$, so (71) holds. $\square$

### A.2.4. LEAKAGE INEQUALITY FOR VECTORS IN THE POPULATION SHARED SUBSPACE

**Lemma A.8** (Projection leakage bound). *For any orthonormal $U, \widehat{U} \in \mathbb{R}^{d \times r}$ and any $x \in \mathrm{span}(U)$,*

$$\|(I - \widehat{U}\widehat{U}^\top)x\|_2 \leq \|(I - \widehat{U}\widehat{U}^\top)U\|_{\mathrm{op}}\,\|x\|_2. \tag{82}$$

*Proof.* Since $x \in \mathrm{span}(U)$, there exists $a \in \mathbb{R}^r$ such that $x = Ua$ and $\|a\|_2 = \|x\|_2$ (because $U^\top U = I_r$). Then

$$\|(I - \widehat{U}\widehat{U}^\top)x\|_2 = \|(I - \widehat{U}\widehat{U}^\top)Ua\|_2 \leq \|(I - \widehat{U}\widehat{U}^\top)U\|_{\mathrm{op}}\,\|a\|_2 = \|(I - \widehat{U}\widehat{U}^\top)U\|_{\mathrm{op}}\,\|x\|_2, \tag{83}$$

which is (82). $\square$

### A.2.5. PROOF OF THEOREM 4.5 AND COROLLARIES

*Proof of Theorem 4.5.* The concentration bound (28) is exactly Lemma A.6, i.e.,

$$\left\|\widehat{C}_t - C_t\right\|_{\mathrm{op}} \ \leq \ \text{RHS of (60)} \tag{84}$$

with probability at least $1 - \delta$.

For the subspace bound, apply Lemma A.7 with $A = C_t$, $\widehat{A} = \widehat{C}_t$, $E = \widehat{C}_t - C_t$, $\gamma = \gamma_t$, $U = U_{\star,t}$, and $\widehat{U} = U_t$. This gives

$$\|(I - U_t U_t^\top)U_{\star,t}\|_{\mathrm{op}} \ \leq \ \min\left\{1, \frac{2}{\gamma_t}\|\widehat{C}_t - C_t\|_{\mathrm{op}}\right\}. \tag{85}$$

By definition (24), the left-hand side equals $\sin\Theta_t$, proving (29). □

*Proof of Corollary 4.6.* Under (30), we have $\sum_{i\in S_t} p_{t,i}^2 = 1/n_{\mathrm{eff}}(t)$ and therefore

$$2B_t^2\sqrt{2\log\left(\frac{2d}{\delta}\right)\sum_{i\in S_t}p_{t,i}^2} \ \leq \ 2B_t^2\sqrt{2\log\left(\frac{2d}{\delta}\right)\cdot\frac{\gamma_t^2\varepsilon^2}{128B_t^4}} \ = \ \frac{\gamma_t\varepsilon}{4}. \tag{86}$$

Similarly, the second constraint in (30) implies

$$\frac{4}{3}B_t^2\, p_{t,\max}\, \log\left(\frac{2d}{\delta}\right) \ \leq \ \frac{4}{3}B_t^2 \cdot \frac{3\gamma_t\varepsilon}{16B_t^2\log\left(\frac{2d}{\delta}\right)} \cdot \log\left(\frac{2d}{\delta}\right) \ = \ \frac{\gamma_t\varepsilon}{4}. \tag{87}$$

Combining (86)–(87) with (28) yields, with probability at least $1 - \delta$,

$$\|\widehat{C}_t - C_t\|_{\mathrm{op}} \ \leq \ \frac{\gamma_t\varepsilon}{2}. \tag{88}$$

Plugging (88) into (29) gives $\sin\Theta_t \leq \frac{2}{\gamma_t} \cdot \frac{\gamma_t\varepsilon}{2} = \varepsilon$. □

*Proof of Corollary 4.7.* Apply Lemma A.8 with $U = U_{\star,t}$ and $\widehat{U} = U_t$ to obtain

$$\|(I - U_t U_t^\top)x\|_2 \ \leq \ \|(I - U_t U_t^\top)U_{\star,t}\|_{\mathrm{op}}\|x\|_2. \tag{89}$$

The right-hand factor equals $\sin\Theta_t$ by definition (24), yielding (31). □

## B. Additional Experimental Details

### B.1. Federated data partitioning and unlearning protocols

**Client partitioning.** For each dataset, we construct IID and non-IID partitions using Dirichlet($\alpha$) label-skew with $\alpha \in \{1.0, 0.5, 0.1\}$. We keep the total number of clients and client selection rate consistent across all compared methods within each dataset/architecture setting.

**Client-level unlearning (main).** We train a global model to convergence with FedAvg. A single target client $u$ is then removed. We compute the *oracle* reference $w^{\setminus u}$ by rerunning the *same* FL schedule (same number of rounds, local epochs, optimizer, client sampling pattern) on $\{D_i\}_{i\neq u}$, and use it only for evaluation.

**Class- and sample-level protocols (for compatibility).** To compare against class-centric baselines (e.g., FUCRT) under a client-removal interface, we use a label-skew partition where the target class(es) are concentrated into a designated subset of clients; class revocation is then implemented as multi-client unlearning by removing those clients. For sample-level unlearning comparisons, we optionally split each physical client into two virtual clients (retain vs. forget shard) so that "sample deletion" becomes removing the forget-shard virtual client; this keeps the unlearning request primitive identical (client removal) while matching the evaluation intent.

### B.2. Metrics and measurement

**Accuracy metrics.** We report Retain/RA (accuracy on retained data), Forget/FA (accuracy on deleted data), and overall test accuracy. For class-unlearning protocols, we additionally report per-class retained metrics when required by baselines (e.g., accuracy/precision on a designated class).

**Retrain fidelity.** We report (i) parameter retrain-gap $\|w^{-u} - w^{\backslash u}\|_2$ and (ii) performance deltas relative to $w^{\backslash u}$ on retain/forget/test splits.

**Privacy (membership inference).** We evaluate privacy leakage after unlearning via membership inference attacks (MIA) using a standard shadow-model attack pipeline; we report attack accuracy (or success rate) where lower is better.

**Efficiency.** We measure (i) unlearning wall-clock time, (ii) total communication (bytes transmitted during unlearning and any post-unlearning rounds), and (iii) additional server storage beyond a standard FedAvg checkpoint. For IDFT, the extra storage is dominated by the influence coefficients $\{z_{T,i}\}_{i=1}^{N} \in \mathbb{R}^{N \times s}$; client bases $\{V_i\}$ are generated deterministically from public seeds and are not stored.

### B.3. Hyperparameters and tuning

**IDFT logging.** We set the shared-subspace rank $r \in \{4, 8, 16\}$ and influence dimension $s \in \{32, 64, 128\}$. Entanglement weights use $a \in [0.3, 0.7]$, $b \in [0.1, 0.3]$ with $a + b \leq 1$, and shrinkage $\lambda \in \{0, 0.5, 1, 5, 10\}$. We select hyperparameters using a held-out validation set by minimizing retrain-gap subject to retaining accuracy.

**Deletion and repair.** For constrained repair, we tune proximal weight $\beta \in \{10^{-2}, 10^{-1}, 1\}$ and repair rounds $R \in \{0, 1, 3, 5, 10\}$. We keep $R \ll T$ in all settings and report the accuracy–efficiency trade-off curves when varying $R$.

### B.4. Details: Removability Visualization

We provide mechanistic evidence that IDFT separates (i) shared covarying directions and (ii) client-specific removable influence, while the remaining discrepancy concentrates in a small missed residual that co-varies with entanglement.

For a selected client $u$, in each round $t$ where $u \in S_t$, we compute:

$$\|\Delta_{t,u}^{\mathrm{sh}}\|_2, \qquad \|\widehat{r}_{t,u}\|_2, \qquad \|r_{t,u} - \widehat{r}_{t,u}\|_2,$$

where the shared component is $\Delta_{t,u}^{\mathrm{sh}} = U_t U_t^\top \Delta_{t,u}$, the residual is $r_{t,u} = (I - U_t U_t^\top)\Delta_{t,u}$, and the stored removable residual is $\widehat{r}_{t,u} = s_{t,u} V_u V_u^\top r_{t,u}$ with shrinkage $s_{t,u} = 1/(1 + \lambda \mathcal{E}_{t,u})$.

To visualize how the update magnitude is allocated, we plot the *per-round proportion* of each component:

$$\rho_{t,u}^{\mathrm{sh}} = \frac{\|\Delta_{t,u}^{\mathrm{sh}}\|_2}{Z_{t,u}}, \quad \rho_{t,u}^{\mathrm{st}} = \frac{\|\widehat{r}_{t,u}\|_2}{Z_{t,u}}, \quad \rho_{t,u}^{\mathrm{miss}} = \frac{\|r_{t,u} - \widehat{r}_{t,u}\|_2}{Z_{t,u}}, \quad Z_{t,u} = \|\Delta_{t,u}^{\mathrm{sh}}\|_2 + \|\widehat{r}_{t,u}\|_2 + \|r_{t,u} - \widehat{r}_{t,u}\|_2. \tag{90}$$

By construction, $\rho_{t,u}^{\mathrm{sh}} + \rho_{t,u}^{\mathrm{st}} + \rho_{t,u}^{\mathrm{miss}} = 1$ for each plotted round.

We pick two representative clients based on cumulative entanglement $\sum_t \mathcal{E}_{t,u}$: one from the lower tail (low-entanglement) and one from the upper tail (high-entanglement). We overlay the normalized entanglement $\widetilde{\mathcal{E}}_{t,u} \in [0,1]$ (min–max normalized across plotted rounds for each client) to highlight synchronization between entanglement and the missed portion.

Fig. 2 offers a mechanistic decomposition consistent with IDFT's design goals: (i) $\rho_{t,u}^{\mathrm{sh}}$ is consistently the largest fraction, indicating $U_t$ captures substantial covarying directions; (ii) $\rho_{t,u}^{\mathrm{st}}$ remains stable and non-trivial, showing that IDFT continuously accumulates a separable influence signal that supports subtraction; and (iii) $\rho_{t,u}^{\mathrm{miss}}$ stays small but increases transiently alongside $\widetilde{\mathcal{E}}_{t,u}$, supporting the view that the missed residual corresponds to harder-to-unlearn (more entangled/leaky) components that are corrected by short repair.

## C. Additional Experimental Results

### C.1. Hyperparameter Sensitivity

We analyze the sensitivity of IDFT to key hyperparameters that may affect (i) *training-time influence logging* and (ii) *deletion-time repair*. We vary one hyperparameter at a time while keeping others fixed at the default configuration

$\{r, s, \lambda, \beta, R\} = \{8, 128, 5, 0.1, 5\}$, and evaluate on CIFAR10-ResNet18 under the label-flipping benchmark (Table 1B) as well as the IID 10-client fidelity metric Avg. Gap (Table 1A). We report mean±std over 3 random seeds.

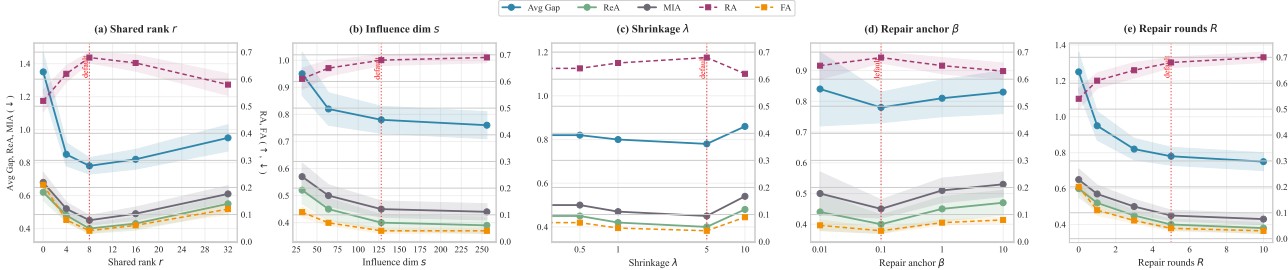

*Figure 3.* **Hyperparameter sensitivity of IDFT (mean±std over 3 seeds).** We sweep one hyperparameter while fixing others at the default $\{r, s, \lambda, \beta, R\} = \{8, 128, 5, 0.1, 5\}$.

Fig. 3 probes the robustness of IDFT along its two core mechanisms: (i) *what gets logged as removable influence* (controlled by $r, s, \lambda$), and (ii) *how aggressively we close the remaining discrepancy* after subtraction (controlled by $\beta, R$). In general, $r$ trades off *filtering cross-client covarying directions* versus leaving more signal in the residual: too small $r$ pushes shared directions into the logged residual, which can amplify subtraction errors and increase retrain gap, while moderate $r$ yields stable fidelity and utility. The influence dimension $s$ mainly controls *capacity* of per-client removable components: increasing $s$ reduces approximation error in the influence bank but typically exhibits diminishing returns once the dominant residual directions are captured. Shrinkage strength $\lambda$ mediates *removability*: $\lambda = 0$ stores (and later subtracts) entangled components, which may hurt retain utility and privacy metrics, whereas overly large $\lambda$ can under-store influence and shift burden to repair; a moderate range is stable. On the repair side, $\beta$ sets the trust region around the subtraction initializer: small $\beta$ allows faster correction toward the leave-out optimum but may require careful control to avoid drift under limited rounds, while large $\beta$ preserves the initializer but may leave residual gap. Finally, increasing $R$ monotonically improve fidelity-related metrics (Avg. Gap/ReA/MIA) with diminishing gains, at the cost of higher unlearning-time compute/communication; in practice, IDFT is intended to operate in the small-$R$ regime. Overall, IDFT is not sensitive to hyperparameters and can achieve superior performance within a relatively wide range.

## C.2. Error Attribution

Our theory (§4.1) suggests that the final retrain gap is driven by the *unremoved* per-round residual $\Delta_{t,u}^{\mathrm{miss}}$ (Eq. (17)) that IDFT fails to subtract. We ask: *Is the empirical gap strongly (near-linearly) correlated with $\sum_t \|\Delta_{t,u}^{\mathrm{miss}}\|_2$ (or the corresponding accumulated bound term)?*

For each deletion request (client $u$) across multiple random seeds and heterogeneity levels (Dirichlet $\alpha$), we compute

$$X_u \triangleq \sum_{t:\, u \in S_t} \left\|\Delta_{t,u}^{\mathrm{miss}}\right\|_2 \qquad \text{and} \qquad Y_u \triangleq \|w^{-u} - w^{\backslash u}\|_2 \;\; (\text{Avg. Gap}), \qquad (91)$$

and visualize the relationship $Y_u$ vs. $X_u$. Each point corresponds to one deletion instance. Point color encodes heterogeneity ($\alpha$), and point size encodes the cumulative entanglement $\sum_t \mathcal{E}_{t,u}$ (Eq. (8)).

Fig. 4 exhibits a strong, near-linear relationship between $X_u = \sum_t \|\Delta_{t,u}^{\mathrm{miss}}\|_2$ and the final gap $Y_u$, supporting the central mechanism claim: *what IDFT fails to remove during subtraction largely determines the retrain discrepancy.* Moreover, more heterogeneous regimes (smaller $\alpha$, darker colors) and higher entanglement (larger markers) tend to concentrate in the upper-right region, consistent with the intuition that non-IID training amplifies cross-client co-adaptation and increases the "hard-to-unlearn" component. Overall, the plot empirically validates that the logging+shrinkage design provides a faithful proxy for deletion difficulty.

## C.3. Pareto Frontier

Do we provide a genuinely better trade-off between unlearning fidelity and unlearning cost, rather than only winning at a single operating point? We visualize a Pareto scatter in the (cost, fidelity) plane. As cost, we use the reported unlearning communication volume under the client-unlearning benchmark in Table 1(B). As fidelity, we use *ReA* ($\downarrow$), where smaller

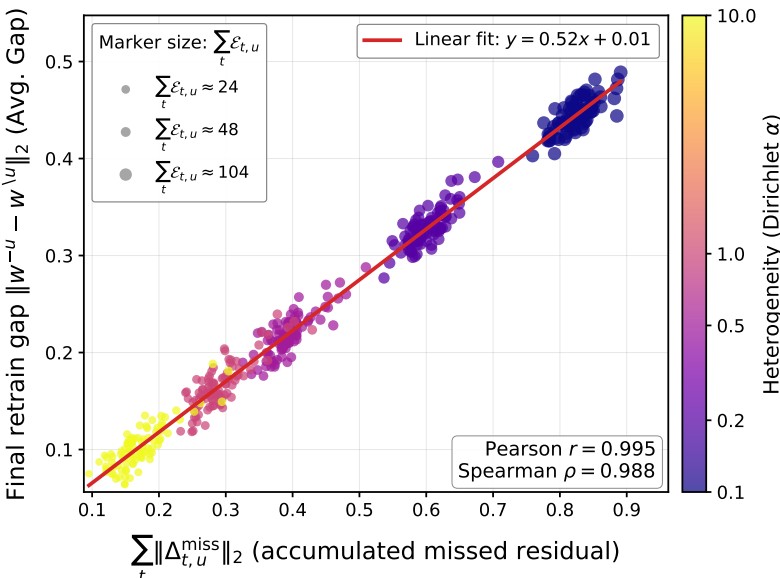

*Figure 4.* **Retrain gap vs. accumulated missed residual.** Each point is one deletion instance. Color indicates Dirichlet heterogeneity $\alpha$ (smaller $\alpha \Rightarrow$ more non-IID); marker size indicates $\sum_t \mathcal{E}_{t,u}$. A fitted line and correlation coefficients quantify the near-linear association predicted by our analysis.

is better. Each baseline corresponds to one point (its reported `Comm` and `ReA`). IDFT produces a *curve* by varying repair rounds $R$: larger $R$ increases cost (more repair communication) but improves fidelity. Fig. 5 highlights the lower envelope (Pareto frontier) among all points.

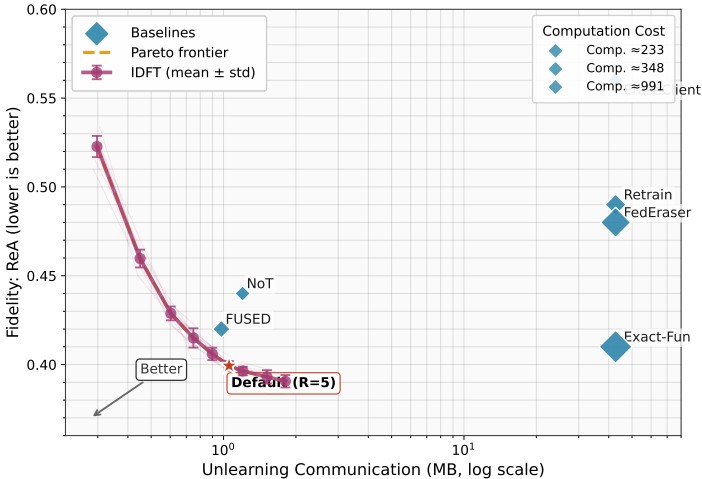

*Figure 5.* **Pareto frontier of fidelity vs. unlearning cost.** X-axis is unlearning communication (log-scale); Y-axis is ReA ($\downarrow$). Baselines are single points; IDFT traces a curve by varying $R$. The dashed line indicates the empirical Pareto frontier (lower-left is better).

Fig. 5 shows that IDFT forms a favorable frontier: increasing $R$ smoothly trades additional communication for improved ReA, and the default operating point (e.g., $R$=5) lands near the lower-left region with strong fidelity at moderate cost. In contrast, history-heavy methods (e.g., FedEraser-style) incur much larger communication without commensurate fidelity gains, while extremely low-cost methods tend to plateau at higher ReA. This plot therefore supports that IDFT improves the *fidelity–cost* frontier rather than only outperforming at a single point.

### C.4. Generalization of the Entanglement Metric Across Tasks and Architectures

Is the entanglement metric $\mathcal{E}_{t,u}$ (Eq. (8)) a principled indicator of deletion difficulty, or a heuristic that only works in a narrow setting? Specifically, do $\mathcal{E}$ and its components $\{\mathcal{E}^{\mathrm{cov}}, \mathcal{E}^{\mathrm{coa}}, \mathcal{E}^{\mathrm{leak}}\}$ (Eqs. (9)–(10)) correlate with deletion errors

$\{\mathrm{Gap}, \mathrm{ReA}\}$ *consistently* across datasets, model architectures, and heterogeneity regimes? For each *setting* (dataset $\times$ architecture $\times$ heterogeneity), we collect many deletion instances (different deleted clients and random seeds). For each instance, we compute (i) the cumulative entanglement scores $\sum_t \mathcal{E}_{t,u}^{(\cdot)}$ and $\sum_t \mathcal{E}_{t,u}$, and (ii) the final deletion errors (Avg. Gap and ReA). We then compute Pearson correlations between each entanglement component and each error metric. Fig. 6 summarizes these correlations as a heatmap.

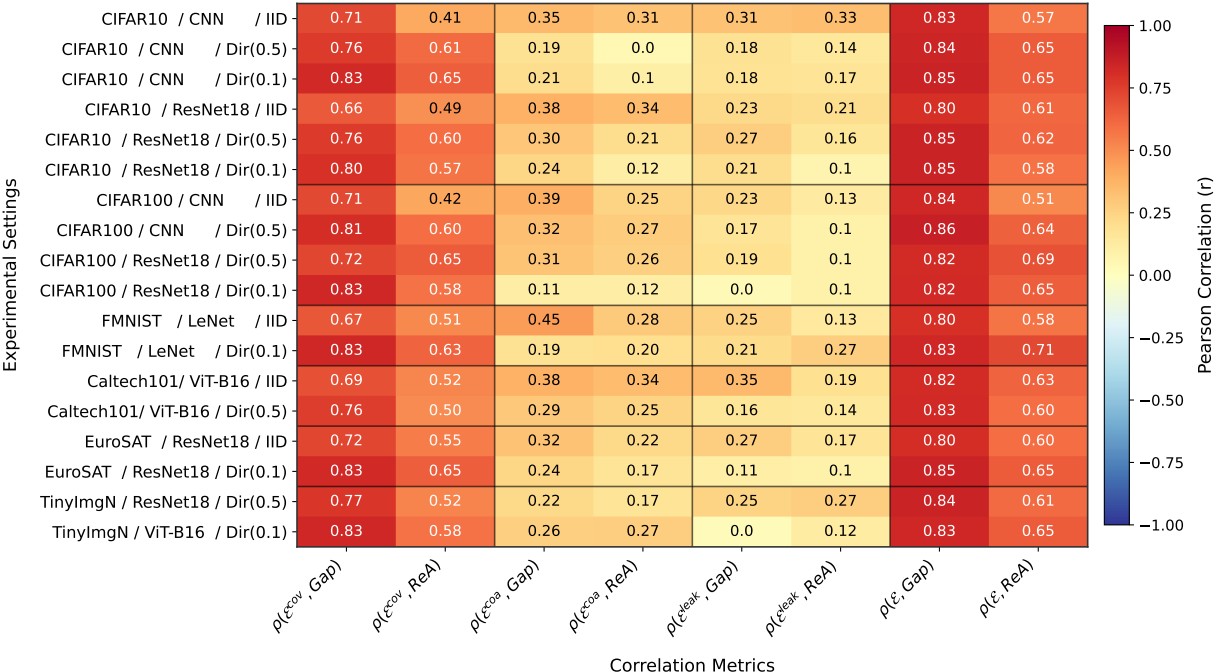

*Figure 6.* **Correlation heatmap of entanglement components vs. deletion errors.** Rows are dataset/architecture/heterogeneity settings; columns are Pearson correlations between $\{\sum_t \mathcal{E}^{\mathrm{cov}}, \sum_t \mathcal{E}^{\mathrm{coa}}, \sum_t \mathcal{E}^{\mathrm{leak}}, \sum_t \mathcal{E}\}$ and $\{\mathrm{Gap}, \mathrm{ReA}\}$. The overall entanglement score $\mathcal{E}$ correlates stably with both errors across settings, while component contributions remain interpretable (e.g., stronger $\mathcal{E}^{\mathrm{cov}}$ in highly non-IID regimes, and stronger $\mathcal{E}^{\mathrm{coa}}$ for deeper architectures).

Fig. 6 validates this diagnostic: the overall entanglement $\mathcal{E}$ maintains a strong positive correlation with both Gap and ReA across datasets and architectures, supporting its role as a *task- and model-agnostic* proxy for deletion difficulty. Meanwhile, the decomposition provides structure: under stronger heterogeneity (smaller Dirichlet $\alpha$), the covariance component $\mathcal{E}^{\mathrm{cov}}$ tends to explain a larger fraction of the error variation (shared directions become harder to disentangle), whereas for deeper backbones (ResNet/ViT), $\mathcal{E}^{\mathrm{coa}}$ becomes more predictive (layerwise co-adaptation spreads influence across blocks). Finally, $\mathcal{E}^{\mathrm{leak}}$ captures settings where the client influence subspace cannot faithfully represent residuals, and thus correlates with larger residual error requiring repair. Together, this analysis supports the view that $\mathcal{E}$ is more than a hand-tuned heuristic: it provides a compact diagnostic that generalizes across regimes and remains interpretable through its components.

